# RoboTransfer: Geometry-Consistent Video Diffusion for Robotic Visual Policy Transfer

## Abstract

The goal of general-purpose robotics is to create agents that can seamlessly adapt to and operate in diverse, unstructured human environments. Imitation learning has become a key paradigm for robotic manipulation, yet collecting large-scale and diverse demonstrations is prohibitively expensive. Simulators provide a cost-effective alternative, but the sim-to-real gap remains a major obstacle to scalability. We present *RoboTransfer*, a diffusion-based video generation framework for synthesizing robotic data. By leveraging cross-view feature interactions and globally consistent 3D geometry, *RoboTransfer* achieves multi-view geometric consistency while enabling fine-grained control over scene elements, including background editing and object replacement. Experiments show that *RoboTransfer* generates videos with improved geometric consistency and visual fidelity, and that policies trained on this data generalize better to novel, unseen scenarios. The code and datasets will be released upon acceptance.

## 1 Introduction

Imitation Learning (IL) has become a fundamental approach for visuomotor control in robotic manipulation (Zhao et al., 2023). However, collecting large-scale real-world robot demonstrations is prohibitively expensive (Brohan et al., 2022; 2023). Simulated environments offer a cost-effective alternative (Xiang et al., 2020; Mu et al., 2025), but the scarcity of assets and sim-to-real gap(Sadeghi et al., 2017; Mehta et al., 2020) make it extremely challenging to scale.

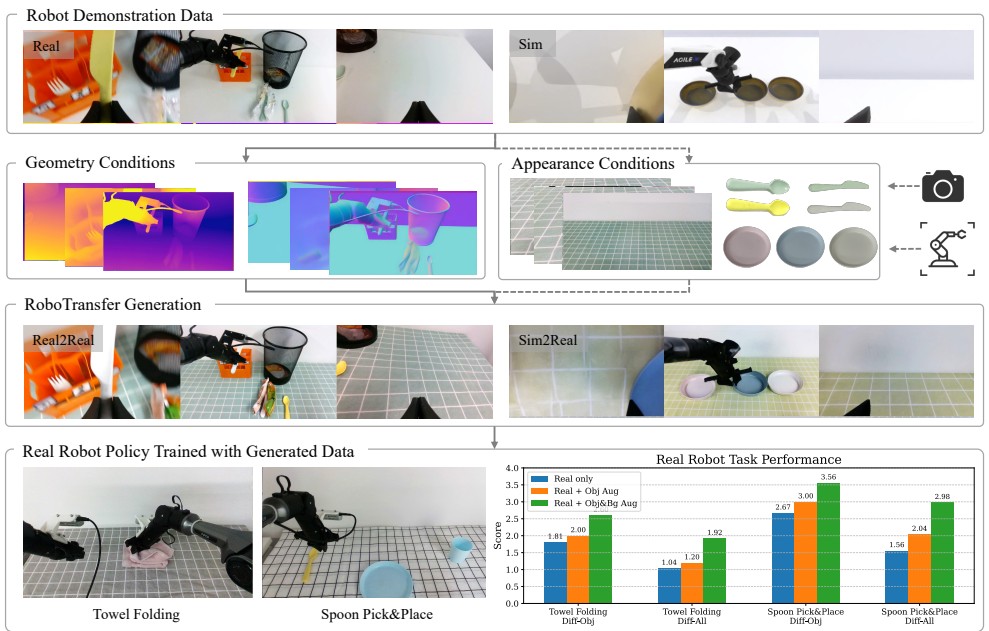

Figure 1: RoboTransfer Overview: Collecting real-world data is expensive, while simulated data often lacks realism. RoboTransfer generates realistic data with multi-view consistency. Experiments demonstrate that the synthesized data enhances real robot policy performance.

Table 1: Comparison of different methods for robotic data generation. RoboTransfer is the only one that provides temporal and multi-view consistency while offering fine-grained control.

| Method | Model Type | Generation Consistency | | Control Level | | |
| --- | --- | --- | --- | --- | --- | --- |
| | | Temporal | Multi-View | Background | Object | Environment |
| ROSIE (Yu et al., 2023) | Image Diffusion | ✗ | ✗ | ✗ | ✔ | ✔ |
| RoboEngine (Yuan et al., 2025) | Image Diffusion | ✗ | ✗ | ✗ | ✔ | ✗ |
| Cosmos-Transfer1 (Alhaija et al., 2025) | Video Diffusion | ✔ | ✗ | ✗ | ✗ | ✔ |
| **RoboTransfer (Ours)** | Video Diffusion | ✔ | ✔ | ✔ | ✔ | ✔ |

Recently, diffusion models (Ho et al., 2020) have gained attention as a promising method to synthetically generate realistic and diverse data. To expand robotic datasets without the need for extensive real-world data collection, ROSIE (Yu et al., 2023) employs text-guided image generation models trained on real-world data. However, it performs image augmentation on a per-frame basis, which leads to a loss of temporal and spatial consistency. In contrast, Cosmos-Transfer (Alhaija et al., 2025) generates photorealistic data using a video diffusion model conditioned on segmentation and depth, which helps preserve geometric consistency.

Despite rapid advancements, two key challenges persist. First, robotic systems often rely on single-view observations, limiting their perception. Multi-view observation is commonly used to address this, but generating consistent multi-view results remains a significant challenge for video generative models. Second, robot manipulation tasks are complex and interactive, and precisely controlling these tasks via textual input alone, as seen in text-to-video frameworks (Blattmann et al., 2023; Zheng et al., 2024), remains a substantial challenge.

To address these challenges, we introduce *RoboTransfer*, a geometry-consistent video diffusion framework tailored for robotic visual policy transfer (Figure 1). As summarized in Table 1, *RoboTransfer* is, to the best of our knowledge, the first framework to guarantee multi-view generation consistency in robotic data synthesis, while also offering fine-grained and disentangled control over both background and object attributes.

The main contributions are as follows:

1. We propose *RoboTransfer*, a video data generation framework for robotic manipulation that ensures multi-view consistency while enabling fine-grained, disentangled control.
2. We introduce a novel data construction pipeline that automatically decomposes real-world robot demonstrations into the geometric and appearance conditions.
3. We demonstrate that *RoboTransfer* generates multi-view videos with substantially improved generation consistency, and that policies trained on this data generalize more effectively to novel, unseen environments.

## 2 RELATED WORK

### 2.1 VISUAL GENERALIZABLE IMITATION LEARNING

Imitation Learning (IL) has become a cornerstone for visuomotor control in robotic manipulation, with deep networks mapping raw visual inputs to motor commands based on demonstrations (Chi et al., 2023; Kim et al., 2024; Zhao et al., 2023; Brohan et al., 2023). However, collecting large-scale task-specific data is prohibitively expensive, leading to the use of auxiliary sources such as human demonstration videos (Grauman et al., 2022; Bi et al., 2025) and operational logs from other platforms (O'Neill et al., 2024; Khazatsky et al., 2024; Bu et al., 2025). These sources introduce domain-specific biases and distributional shifts, which can degrade IL policy performance when transferred to new environments. While simulators can generate large quantities of labeled frames (Todorov et al., 2012; Xiang et al., 2020; Geng et al., 2025; Mu et al., 2025), discrepancies in physics modeling, rendering fidelity, and scene composition hinder sim-to-real transfer. Domain randomization attempts to close this gap (Laskin et al., 2020; Hansen et al., 2020; Kostrikov et al., 2020; Akkaya et al., 2019), but it typically only applies color perturbations across the entire image, without capturing localized or structurally meaningful variations. In this work, we leverage a generative model to synthesize photorealistic multi-view videos for manipulations, significantly enhancing policy robustness and enabling seamless transfer to novel environments.

## 2.2 GENERATION MODELS IN EMBODIED AI

With the rapid advancement of generative models (Kong et al., 2024; Wang et al., 2025a; Blattmann et al., 2023; Alhaija et al., 2025; Yang et al., 2024; Zheng et al., 2024; Fu et al., 2024; Zhang et al., 2025; Li et al., 2025; He et al., 2025), there is a growing trend toward leveraging generated data to bridge the gap between synthetic environments and real-world physical applications (Zhu et al., 2024). In particular, domains like autonomous driving have seen increasing adoption of generative techniques, as demonstrated by (Wang et al., 2024a; Zhao et al., 2025b; Hu et al., 2023; Wang et al., 2024b; Gao et al., 2023; 2024; Zhao et al., 2024a; Ni et al., 2024; Zhao et al., 2025a). Similarly, in robotic scenarios, where real-world data collection is often prohibitively expensive and time-consuming, generative data has proven to be highly beneficial. Models such as UniSim (Yang et al., 2023), UniPi (Du et al., 2023), and RoboDreamer (Zhou et al., 2024) synthesize future robot behaviors via text prompts, which are then translated into actionable commands using inverse dynamics models. However, purely video-based generation methods often suffer from spatial and temporal inconsistencies, leading to degraded performance in downstream action prediction and planning. To address this, TesserAct (Zhen et al., 2025) proposes a multi-modal generation pipeline that simultaneously synthesizes RGB, depth, and normal videos, enabling the reconstruction of coherent 4D scenes. This framework ensures both spatial and temporal consistency in robotic environments and supports policy learning that significantly surpasses prior video-only world models. Moreover, data generalization remains a critical challenge in robot learning. Techniques like ReBot (Fang et al., 2025) and RoboEngine (Yuan et al., 2025) adopt background inpainting to diversify environmental textures, while Cosmos-Transfer (Alhaija et al., 2025) utilizes video-to-video translation to enrich overall scene appearance. Despite their success, these approaches lack fine-grained control over foreground and background textures, often resulting in synthetic data distributions misaligned with real-world tasks, especially when training policy models. Additionally, their inability to ensure multi-view consistency limits their scalability and effectiveness in complex, robotic settings.

## 3 METHODS

We introduce *RoboTransfer*, a framework designed for the controllable generation of multi-view videos to support the training of policy models. By providing explicit control over both scene geometry and appearance, *RoboTransfer* enables the synthesis of video data with precisely defined distributions. This fine-grained control allows for the generation of diverse training scenarios, which are critical for improving the generalization of policy models. In the remainder of this section, we first present the preliminaries of video diffusion models in Sec. 3.1. Then, we describe the framework of *RoboTransfer* in Sec. 3.2. Finally, the dataset construction pipeline is elaborated in Sec. 3.3.

### 3.1 VIDEO DIFFUSION MODEL PRELIMINARIES

Diffusion-based methods for controllable video generation model the process as a gradual refinement of a noise latent $\boldsymbol{\epsilon}$ into a clean video latent $\mathbf{x_0}$, under the guidance of spatially aligned conditions $y_s$ (e.g., depth map) and unstructured conditions $y_u$ (e.g., CLIP embedding). Popular approaches in this paradigm include Flow Matching (Lipman et al., 2022), DDPM (Ho et al., 2020), and EDM (Karras et al., 2022). In the case of EDM, the learning objective is formulated as:

$$\mathcal{L}(D_\theta, \sigma) = \mathbb{E}_{\mathbf{x_0}, y_s, y_u} \left[ \|\mathbf{x_0} - D_\theta \left( \mathcal{E}(\mathbf{x_0} + \mathbf{n}), \tau_u(y_u), \tau_s(y_s), \sigma \right) \|_2^2 \right]. \tag{1}$$

Here, $\mathbf{x_0} \sim p_{\text{data}}$ denotes a clean sample drawn from the dataset, and $\mathbf{n} \sim \mathcal{N}(\mathbf{0}, \sigma^2\mathbf{I})$ represents Gaussian noise. $\mathcal{E}$ denotes the encoder component of a VAE (Kingma et al., 2013), and the denoising network $D_\theta$ is conditioned on the noise level $\sigma$ and encoded conditions $\tau_u(y_u), \tau_s(y_s)$. To encourage stable learning across varying noise magnitudes, the total training objective aggregates the per-noise loss via a weighted expectation:

$$\mathcal{L}(D_\theta) = \mathbb{E}_\sigma \left[ \frac{\lambda(\sigma)}{\exp(u(\sigma))} \mathcal{L}(D_\theta, \sigma) + u(\sigma) \right], \tag{2}$$

$$\lambda(\sigma) = \frac{\sigma^2 + \sigma_{\text{data}}^2}{(\sigma \cdot \sigma_{\text{data}})^2}, \tag{3}$$

$$\ln(\sigma) \sim \mathcal{N}(P_{\text{mean}}, P_{\text{std}}^2). \tag{4}$$

Figure 2: The *RoboTransfer* framework performs multi-view consistent modeling to jointly reason across viewpoints. It represents geometry with metric depth and normal maps, and encodes appearance using reference backgrounds and object-specific images for detailed control over appearance.

In this setup, $\sigma_{\text{data}}$ denotes the empirical standard deviation of the data, while the distribution of $\sigma$ is governed by the hyperparameters $P_{\text{mean}}$ and $P_{\text{std}}$. The weighting function $\lambda(\sigma)$ ensures that all noise levels contribute proportionally during the initial stages of training.

## 3.2 *RoboTransfer* FRAMEWORK

The overall framework of *RoboTransfer* is illustrated in Figure 2. To ensure multi-view consistency during generation, we perform multi-view consistent modeling, enabling the generation process to reason jointly over information from different viewpoints. On the conditions side, *RoboTransfer* incorporates fine-grained control by encoding both geometric and appearance information. Specifically, we represent geometry using metric depth maps and surface normal maps, capturing the underlying 3D structure of the scene. Meanwhile, the appearance is encoded using reference background images and object-specific images, providing detailed control over texture, color, and contextual appearance. In the following sections, we first introduce the multi-view consistent modeling and then describe the encoding mechanisms for geometric and appearance conditions, respectively.

**Multi-view Consistent Modeling.** Robotic manipulation often relies on multi-view camera setups to capture a scene in parallel. To generate multi-view consistent videos for downstream tasks, *RoboTransfer* leverages the multi-view in-context learning capabilities (Zhao et al., 2025b; Huang et al., 2024) inherently present in pretrained diffusion models. Specifically, given $N$ synchronized videos $\{V_1, V_2, ..., V_N\}$ from different viewpoints, we concatenate them along the width dimension and encode the joint information using a Variational Autoencoder(VAE) encoder $\mathcal{E}$:

$$\mathbf{x}_0 = \mathcal{E}([V_1, V_2, \ldots, V_N]), \tag{5}$$

where $[V_1, V_2, \ldots, V_N]$ represents the concatenated multi-view video sequence. This modeling approach leverages the spatial reasoning capabilities of existing video diffusion backbones, requiring no structural modifications, while enabling fast convergence and high-quality, view-consistent video generation in robotic manipulation settings.

**Geometry Conditions Injection.** Recent video generation models primarily learn spatiotemporal coherence through data-driven approaches such as temporal attention (Blattmann et al., 2023; Zheng et al., 2024) or full attention (Wang et al., 2025a; Kong et al., 2024; Alhaija et al., 2025). While effective at capturing pixel-level dynamics, these methods lack an explicit understanding of underlying 3D geometry, limiting their applicability in robotic environments. To address this limitation, *RoboTransfer* explicitly incorporates geometry conditions to enhance the model's awareness of scene structure and depth continuity. Specifically, we utilize depth and surface normal videos as geometric conditions. These two types of geometric cues are complementary; depth maps provide information about spatial distances from the camera, while normal maps encode local surface orientations. When combined, they offer a richer and more holistic description of scene geometry (see Sec. 3.3 for details on data acquisition). Since the depth and surface normal sequences are spatially aligned with their corresponding RGB views, we leverage the VAE encoder composed of stacked convolutional layers that jointly downsample and encode geometric cues from depth and normal videos. The resulting geometry-aware representation is concatenated with the noise latents along the channel dimension, enabling diffusion-based video generation to be precisely guided by consistent and physically plausible 3D cues throughout the generation process.

Figure 3: *RoboTransfer* data construction pipeline generates image pairs with Geometry and Appearance conditions. Geometry conditions (left) with metric depth and normals from demonstration videos and appearance conditions (right) from keyframes. A VLM-based descriptor generates object descriptions, which are processed by Grounding-SAM to create per-object masks.

**Appearance Conditions Injection.** To enable fine-grained control over the generated textures, *RoboTransfer* introduces texture conditions from two complementary perspectives: background appearance and object appearance. Specifically, we use a background reference image and a set of object reference images as condition signals to guide the generation process toward faithful reproduction of scene textures and object-level details. A key challenge in appearance control is ensuring compatibility with the geometry conditions. Naïvely combining appearance and geometry inputs may introduce conflicts (e.g., mismatched depth and texture), thereby weakening both the geometry consistency and the visual fidelity of generated results. To address this, we carefully curate the appearance reference images, resulting in a background reference image $C_b$ and object reference images $C_o$ that do not contradict the geometric priors (see Sec. 3.3 for details). For background appearance, we employ a VAE encoder to compress $C_b$ into a latent representation that is spatially aligned with the generation latents. This encoded background appearance is then concatenated with the latent inputs to control the global texture and background style of the output video. In contrast, object appearance presents additional challenges due to the variable number and spatial distribution of objects in the scene. Therefore, object images are treated as unstructured conditions. We encode each object's appearance using CLIP (Radford et al., 2021), which produces a global embedding used to guide the generation via cross attention. This design allows flexible and scalable conditioning on diverse object appearances while preserving compatibility with the structured geometry inputs.

### 3.3 DATASET CONSTRUCTION FROM REAL ROBOTIC DEMONSTRATION DATA

Data construction is central to our framework, where real-world recording data is decomposed into high-quality triplets for training *Robotransfer*, consisting of Geometry Conditions, Appearance Conditions, and ground-truth images. The overall pipeline is illustrated in Figure 3.

**Geometry Conditions Construction** Starting from robotic demonstration data with two wrist cameras and one head camera. To enforce spatial and temporal consistency, we incorporate geometric cues such as depth and surface normals. Since some robots only have RGB sensors and raw depth maps from RGB-D sensors are often noisy, we use a state-of-the-art depth estimator (Chen et al., 2025; Wang et al., 2025b) to produce consistent depth maps. For unmetric depth outputs, we align the estimated scale with the RGB-D sensor using robust least-squares fitting to ensure global spatial accuracy. For surface normals, we compute per-frame estimates using state-of-the-art monocular normal estimators (He et al., 2024; Wang et al., 2025b), offering conditions for detailed geometry.

**Appearance Conditions Construction.** To enable controllable data generation with the diffusion model, we sample multiview RGB keyframes as appearance conditions. A VLM-based descriptor generates scene and object descriptions, which guide the Grounding-SAM module (Ren et al., 2024) to detect and segment per-object masks. To obtain clean background references, the segmented objects are inpainted in the original images, producing plausible object-free scenes (e.g., an empty tabletop). For object-level conditions, the segmented masks are further processed with the CLIP model (Radford et al., 2021) to extract semantic embeddings.

Table 2: Experiment comparison on different geometry conditions of *RoboTransfer*: It shows that combining metric depth and normal maps yields the best consistency across all views and metrics.

| Model | Camera | RMSE ↓ | Abs.Rel. ↓ | Sq.Rel. ↓ | Mean Err. ↓ | Med.Err. ↓ | Pix.Mat. ↑ | FVD ↓ |
|---|---|---|---|---|---|---|---|---|
| *RoboTransfer* [D.S.] | left | 0.074 | 0.124 | 0.020 | 4.86 | 2.88 | 142.90 | 218.51 |
| *RoboTransfer* [D.P.] | left | 0.054 | 0.090 | 0.010 | 3.91 | 2.28 | 149.68 | 123.31 |
| *RoboTransfer* [Metric D.P.] | left | 0.049 | 0.081 | **0.008** | 3.48 | 1.99 | 183.26 | 112.43 |
| *RoboTransfer* [Metric D.P. + N.] | left | **0.047** | **0.079** | **0.008** | **3.31** | **1.92** | **202.03** | **107.43** |
| *RoboTransfer* [D.S.] | head | 0.182 | 0.074 | 0.031 | 3.51 | 1.58 | – | 153.76 |
| *RoboTransfer* [D.P.] | head | **0.132** | **0.053** | **0.015** | 3.05 | 1.43 | – | **95.89** |
| *RoboTransfer* [Metric D.P.] | head | 0.134 | 0.054 | **0.015** | 2.93 | 1.40 | – | 103.32 |
| *RoboTransfer* [Metric D.P. + N.] | head | 0.133 | 0.054 | **0.015** | **2.86** | **1.39** | – | 101.17 |
| *RoboTransfer* [D.S.] | right | 0.090 | 0.137 | 0.025 | 4.93 | 2.91 | 40.70 | 396.33 |
| *RoboTransfer* [D.P.] | right | 0.072 | 0.103 | 0.016 | 4.14 | 2.36 | 56.02 | 262.96 |
| *RoboTransfer* [Metric D.P.] | right | 0.064 | 0.090 | 0.011 | 3.74 | 2.11 | 65.45 | 226.76 |
| *RoboTransfer* [Metric D.P. + N.] | right | **0.058** | **0.087** | **0.009** | **3.55** | **2.00** | **75.67** | **220.12** |

## 4 EXPERIMENTS FOR SYNTHESIS QUALITY

To evaluate the synthesis quality of *RoboTransfer*, including multi-view geometric consistency and controllability, we perform both quantitative and qualitative analyses. In this section, we first present the evaluation metrics and analysis of the synthesis quantity in Sec. 4.1. Then, the synthesis quality results are presented in Sec. 4.2.

### 4.1 SYNTHESIS QUANTITATIVE ANALYSIS

**Implementation Details.** *RoboTransfer* is fine-tuned from the pre-trained SVD (Blattmann et al., 2023) model. During inference, we use the EDM scheduler (Karras et al., 2022) to perform 30 denoising steps and apply classifier-free guidance. More details are provided in the appendix.

**Evaluation Metrics.** We evaluate the generated videos from multiple perspectives, including multi-view consistency, geometric consistency, and semantic consistency. For multi-view consistency, we follow (Bai et al., 2024) and utilize the state-of-the-art image matching method (Shen et al., 2024) to compute the number of matched pixels (Mat.Pix.) between adjacent views (i.e., left-to-center and right-to-center). For geometric consistency, we adopt the evaluation framework from Cosmos-Transfer (Alhaija et al., 2025), which assesses depth and surface normal alignment. For depth prediction, we compute scale-invariant metrics including Root Mean Squared Error (RMSE), Absolute Relative Error (Abs.Rel.), and Squared Relative Error (Sq.Rel.). For normal estimation, we report the Mean Angular Error (Mean Err.) and Median Angular Error (Med. Err.). In terms of semantic consistency, we measure appearance controllability using CLIP (Radford et al., 2021) similarity. For background-level similarity, we compute the CLIP cosine similarity between a reference background image and each generated frame (BG. Sim.). For foreground object similarity, we use GroundingDINO (Liu et al., 2024a) and SAM2 (Ravi et al., 2024) to segment objects in each generated frame and compute their CLIP similarity with a reference image (Obj. Sim.). Due to motion and occlusion in the wrist-mounted (left and right) views, object tracking becomes unreliable. Therefore, we evaluate this metric only on the center view to ensure accuracy. All the above metrics are computed on a per-frame basis, and we report the average value across all frames as the final score. Additionally, we compute the FVD (Unterthiner et al., 2019) between the generated and real videos to further assess the overall realism and temporal coherence of the generated data.

**Geometry Consistency Analysis.** We first validate the geometric consistency in *RoboTransfer*'s generative capability. As shown in Table 2, directly using raw depth sensor data (D.S.) as a conditional input introduces considerable noise, which adversely affects the generation quality. This degradation is evident across all three views in terms of depth accuracy, surface normals, and multi-view consistency. In contrast, employing model-predicted (Chen et al., 2025) depth (D.P.) results in smoother depth maps, which serve as a more stable supervisory signal for training the generative model. Consequently, the generated videos exhibit significantly better geometric consistency. Specifically, compared to D.S.-based generation, D.P.-based conditions relatively improve the RMSE and Mean Err. of the middle view by 27.4% and 14.2%, and increase Pix.Mat. of the middle and left/right views by 4.7% and 37.6%, respectively. Further improvements are observed when we incorporate metric predicted depths (Metric D.P.) to enforce a consistent depth scale across multiple views. This multi-view scale alignment further enhances geometric consistency, leading to relative improvements of 22.4% and 16.8% in Pix.Mat. of left and right views. The Pix.Mat. is omitted for the head camera since it

Table 3: Experiment comparison on different appearance conditions of *RoboTransfer*.

| Bg. Inpaint | Obj. Split | Camera | BG. Sim. ↑ | Obj. Sim. ↑ | RMSE ↓ | Med.Err. ↓ | Pix.Mat. ↑ | FVD ↓ |
|:---:|:---:|:---:|:---:|:---:|:---:|:---:|:---:|:---:|
| | | left | 0.796 | – | 0.051 | 1.94 | 191.59 | 117.25 |
| ✓ | | left | 0.802 | – | 0.049 | 1.92 | 198.32 | 119.65 |
| | ✓ | left | 0.797 | – | 0.050 | 1.93 | 196.85 | 118.22 |
| ✓ | ✓ | left | **0.805** | – | **0.047** | **1.92** | **202.03** | **107.43** |
| | | head | 0.712 | 0.847 | 0.137 | 1.46 | – | 108.65 |
| ✓ | | head | 0.719 | 0.845 | 0.135 | 1.41 | – | 105.66 |
| | ✓ | head | 0.712 | 0.855 | 0.136 | 1.42 | – | 105.54 |
| ✓ | ✓ | head | **0.720** | **0.858** | **0.133** | **1.39** | – | **101.17** |
| | | right | 0.753 | – | 0.064 | 2.12 | 71.71 | 226.84 |
| ✓ | | right | 0.762 | – | 0.060 | 2.04 | 72.44 | 223.91 |
| | ✓ | right | 0.754 | – | 0.061 | 2.05 | 72.68 | 222.11 |
| ✓ | ✓ | right | **0.764** | – | **0.059** | **2.00** | **75.67** | **220.12** |

remains static. Finally, combining normal and depth as joint conditions (Metric D.P.+N.) provides complementary geometric cues, yielding the best overall performance across all geometric metrics.

**Appearance Consistency Analysis.** We then evaluate *RoboTransfer*'s capability in controlling visual appearance. As shown in Table 3 (row-1 and row-2 of all three camera views), we observe that directly conditioning on background images without inpainting can corrupt the geometric cues. In contrast, inpainting the background before conditioning not only preserves the geometric structure but also improves appearance consistency, leading to $\sim 1\%$ relative improvements in RMSE, median error, and background similarity. Comparing row-1 and row-3 of the head camera view, we examine object-level appearance control. In row-1, the entire image with all the objects is encoded using a global CLIP feature. In contrast, row-3 applies a finer-grained approach by masking and encoding each object individually, followed by feature concatenation. These object-wise conditions lead to a 1% improvement in object CLIP similarity, indicating better control over individual object appearances. Finally, row-4 of all three camera views demonstrates that combining background inpainting with object-wise CLIP encoding achieves the best performance in both foreground and background appearance control. This setup yields the highest CLIP similarity scores for both regions, confirming the effectiveness of jointly modeling structured and unstructured visual components.

## 4.2 SYNTHESIS QUALITATIVE RESULT

**Real-to-Real Synthesis.** We demonstrate that *RoboTransfer* enables controllable manipulation of both foreground and background appearance while preserving geometric and multi-view consistency across camera views. As shown in Figure 4, *RoboTransfer* successfully modifies the background appearance while retaining the foreground object's texture and scene geometry. Figure 5 illustrates that the model can alter the foreground appearance without affecting the background or spatial layout. Finally, Figure 6 demonstrates the model's ability to jointly edit both foreground and background elements. All visualizations confirm that *RoboTransfer* maintains multi-view consistency throughout the generated scenes. More qualitative results are provided in the appendix.

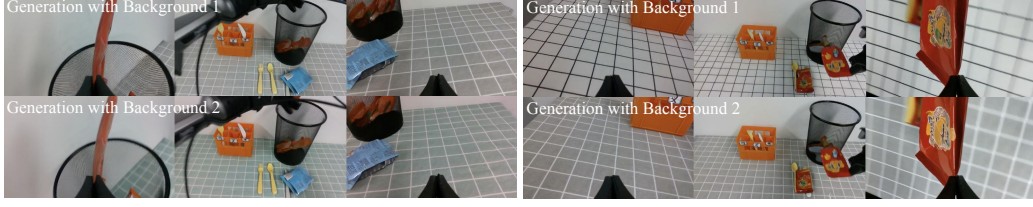

Figure 4: Visualizations of *RoboTransfer* with different background reference images.

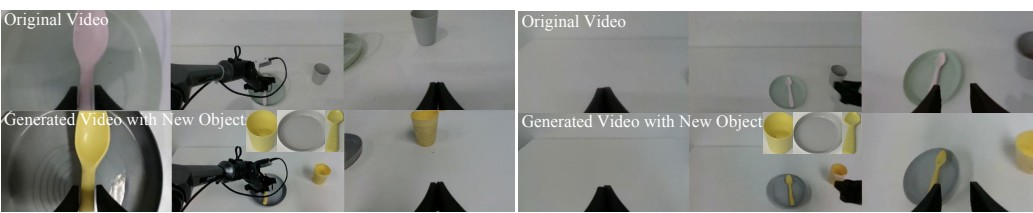

Figure 5: Visualizations of *RoboTransfer* with different object reference images.

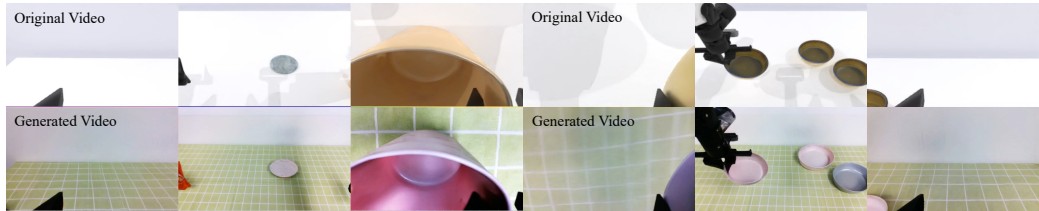

Figure 6: Visualizations of *RoboTransfer* with different background and object reference images.

**Sim-to-Real Synthesis** In simulation, per-view geometry conditions can be obtained at no additional cost. As shown in Figure 7, *RoboTransfer* generates photorealistic videos from simulated geometry inputs, including out-of-distribution cases. This Sim-to-Real paradigm reduces reliance on real-world geometry data, thereby better supporting downstream robotic learning tasks.

Figure 7: Visualizations of *RoboTransfer* Sim-to-Real generation results.

## 5 REAL ROBOT EXPERIMENTS

To evaluate the effectiveness of *RoboTransfer*, we conducted experiments on a real-world robot to assess how synthetic data impacts visual policy generalization across diverse environments. In the remainder of this section, we first describe the robot experimental platform and implementation in Sec. 5.1, followed by an analysis of synthetic data effectiveness and data proportions for enhancing real robot policy robustness in Sec. 5.2.

### 5.1 EXPERIMENTAL PLATFORM AND IMPLEMENTATION

**Experimental Platform and Setup Details.** Our experiments were conducted on the Agilex Cobot Magic platform, which is equipped with two PIPER robotic arms and three Intel RealSense D435i cameras, two mounted on the wrists and one positioned overhead. Only RGB data was used for policy learning, presenting a challenging visual understanding problem.

**Implementation Details.** We adopted ACT (Zhao et al., 2023) as our baseline architecture without modifications. For each task, we collected 100 expert demonstrations via ALOHA (Zhao et al., 2024b) teleoperation. To demonstrate generalizability, the scenes and tasks differ from those in the *RoboTransfer* training data (see Sec. 4 for details). More details are provided in the appendix.

### 5.2 EFFECTIVENESS OF SYNTHETIC DATA

**Benchmark Task and Evaluation Methodology.** We evaluate synthetic data on two long-horizon dual-arm manipulation tasks: spoon pick-and-place and towel folding. Spoon pick-and-place consists of four phases: the left arm grasps a spoon, places it on a tray, the right arm grasps it, and places it in a cup. Towel folding includes three stages: grasping the bottom corners, lifting and folding upward, and folding the right side to the left. The towel task provides a rigorous test of physical plausibility and temporal coherence in complex manipulations. We introduce a Stage Score to measure phase completion, offering finer-grained insight than binary success metrics and enabling evaluation of policy progression and generalization. Experiments were conducted under two conditions: novel objects (**Diff-Obj**), and variations in both objects and environment (**Diff-All**). Testing protocols are detailed in the supplementary material.

**Effect of Synthetic Data Proportions.** To identify the optimal data mixture, we evaluated the performance while varying the proportion of synthetic data from 0% to 100% on the spoon pick and place task under **Diff-All** conditions. As shown in Figure 8, both metrics peak at a 50/50 ratio.

At this balance, the success rate rises from 13.3% to 46.7%, and the stage score nearly doubles from 1.6 to 3.0. Increasing the synthetic proportion further yields diminishing returns, as synthetic data—despite being geometrically conditioned—can lack subtle physical plausibility (e.g., contact dynamics or material properties), which becomes detrimental at very high ratios. A policy trained purely on synthetic data still achieves a 40.0% success rate, outperforming the real-only baseline. This highlights the complementary roles of the two data sources: synthetic data provides visual diversity, while real data grounds the policy in real-world physics. Accordingly, we adopt the 50/50 ratio in the Effectiveness of Synthetic Data Analysis.

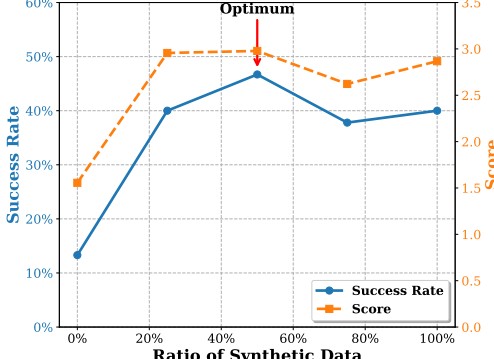

Figure 8: Performance across synthetic data mixing ratios. Note that 0% represents our baseline model trained only on real data.

**Effectiveness of Synthetic Data Analysis.** As shown in Table 4, our synthetic data augmentation significantly enhances policy robustness where the baseline fails. For the Spoon Pick and Place task under the most difficult **Diff-All** condition, our approach achieves a 251% relative improvement in success rate (from 13.3% to 46.7%). The results also reveal a clear cumulative benefit of our augmentation strategy. In the Towel Folding **Diff-Obj** setting, for instance, the Stage Score increases from 1.81 (real only) to 2.0 with object augmentation, and further to 2.6 after adding background augmentation. This confirms that augmenting both object and background appearance is key to learning policies that can handle significant visual domain shifts.

Table 4: Effectiveness of Synthetic Data Augmentation. Performance comparison across different data augmentation strategies. Success Rate (%) and Score are reported.

| Data Composition | Spoon Pick&Place | | | | Towel Folding | | | |
| --- | --- | --- | --- | --- | --- | --- | --- | --- |
| | Diff-Obj | | Diff-All | | Diff-Obj | | Diff-All | |
| | SR | Score | SR | Score | SR | Score | SR | Score |
| Real only | 33.3% | 2.67 | 13.3% | 1.56 | 16.7% | 1.81 | 12% | 1.08 |
| Real + Obj Aug | 44.4% | 3.00 | 22.2% | 2.04 | 16.7% | 2.00 | 12% | 1.24 |
| Real + Obj&Bg Aug | 66.7% | 3.56 | 46.7% | 2.98 | 50.0% | 2.60 | 28% | 1.92 |

## 6 CONCLUSION

In this work, we proposed *RoboTransfer*, a diffusion-based data synthesis framework for robotics that integrates multi-view geometry while providing explicit control over background and object attributes. We also introduced a dataset construction pipeline that generates high-quality triplets incorporating both global geometry and appearance conditions. Experimental results demonstrate that *RoboTransfer* produces multi-view consistent data, significantly improving the generalization of visuomotor policies for robotic manipulation.

**Limitations and Future Work.** While general and flexible, our method currently focuses on augmenting object and scene appearance rather than generating new motions. Future work will explore integration with additional simulators to broaden applicability and optimize the generation architecture for interactive, real-time data synthesis.

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

# Appendix

## A    ROBOTRANSFER IMPLEMENTATION DETAILS

### A.1    TRAINING DETAILS

**Training Dataset.** To construct our training dataset, we leverage the open-source Cobot Magic platform to collect a large-scale video corpus of dual-arm robot executions(see Sec. B.1 for details). The raw videos are segmented into 10Hz clips of 30 frames each, resulting in approximately 24k clips for training. Additionally, we curate a set of 1.6k 10Hz 30-frame clips from the collected dataset for video synthesis quality evaluation. All videos are annotated with conditions as described in Sec. 3.3, to facilitate both training and evaluation.

**Training Details.** *RoboTransfer* is fine-tuned from the pre-trained Stable Video Diffusion (Blattmann et al., 2023) model. During training, videos from each camera view are resized to a resolution of $640 \times 384$. We adopt the AdamW(Loshchilov & Hutter, 2017) optimizer with a learning rate of $3 \times 10^{-5}$ and a global batch size of 8, training for a total of 70K steps. During inference, we use the EDM scheduler (Karras et al., 2022) to perform 30 denoising steps and apply classifier-free guidance.

### A.2    MODELING AND CONDITIONS INJECTION DETAILS

**Object Condition.** We resize each object image to $224 \times 224$ and pass it through CLIP's image encoder to obtain a single global feature vector per object. These embeddings are then concatenated and fed into the diffusion model. This design allows for finer-grained control, enabling individual manipulation of each object.

**Multi-View Consistency Modeling.** Previous methods typically introduce a cross-view module to enhance consistency between views. In contrast, *RoboTransfer* simply concatenates multi-view images, integrating inter-view consistency into global spatial consistency, thereby improving multi-view video modeling. Moreover, this design allows direct loading of pre-trained single-view video generation model weights, without requiring significant modifications to the model architecture.

## B    ROBOTRANSFER DATASET CONSTRUCTION DETAILS

For training *RoboTransfer*, we collected a dedicated dataset and designed a construction pipeline (Figure 3). Here, we provide additional implementation details.

### B.1    DATA COLLECTION

**Robot Platform.** We built a large-scale robotic demonstration dataset using the Agilex Cobot Magic platform, following standardized protocols (Mu et al., 2025; Liu et al., 2024b). Each demonstration includes synchronized RGB-D streams from three Intel RealSense D435i cameras: two hand-eye views and one overhead view (Figure 9).

**Dataset Design.** To capture diverse manipulation scenarios, we designed twelve distinct tasks (Figure 10) with variations in objects, backgrounds, and interactions. For each task, 100 demonstration segments were collected across 10 unique object configurations, totaling 1,000 samples per task. Backgrounds range from textured tabletops to cluttered surfaces, and objects vary in shape, size, and material. This ensures high diversity and realism for robust visual policy training. To further enhance background diversity, we incorporate the AgiBot-World dataset (Bu et al., 2025), which is excluded from ablation studies in Sec. 4.1.

### B.2    GEOMETRY CONDITIONS CONSTRUCTION

**Depth Conditions.** For RGB-D data, raw sensor depth is often noisy or incomplete, particularly on low-reflectivity surfaces, causing domain gaps across samples. Simulator-rendered depth is complete and noise-free. Existing depth completion methods often lack temporal consistency. We adopt Video Depth Anything (VAD) (Chen et al., 2025) to produce temporally coherent, spatially complete depth maps. VAD predictions, however, lack global scale consistency. We align predicted depths to RGB-D sensor measurements using a robust multi-frame least-squares fitting strategy (Figure 11) that iteratively filters outliers to ensure accurate global scale recovery. For datasets without multi-view

Figure 9: Robot platform visualizations.

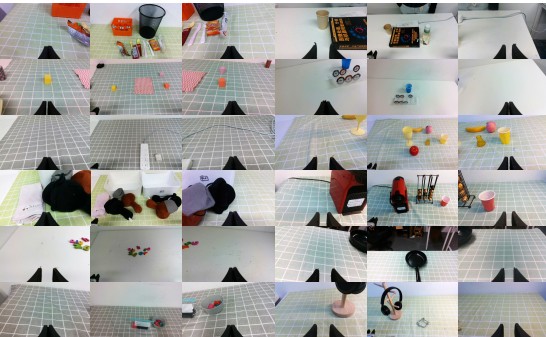

Figure 10: Dataset collected with the Agilex robot.

RGB-D sensors, such as AgiBot-World (Bu et al., 2025), we estimate metric depth using MoGe (Wang et al., 2025b).

---

**Algorithm 1** Dynamic Mask Alignment

**Require:** $\mathbf{D}_{\text{pred}}, \mathbf{D}_{\text{sensor}} \in \mathbb{R}^{B \times H \times W}$
**Ensure:** $\mathbf{D}_{\text{Metric}} \in \mathbb{R}^{B \times H \times W}$
1: **Initialize mask:**
$$\mathcal{M} \leftarrow (\mathbf{D}_{\text{sensor}} > \epsilon) \wedge (\mathbf{D}_{\text{pred}} > \epsilon)$$
2: **for** 2 iterations **do**
3:     **Scale Fitting:**
4:     $s, b \leftarrow$ Scale Fitting$(\mathbf{D}_{\text{pred}}, \mathbf{D}_{\text{sensor}}, \mathcal{M})$
5:     $\mathbf{D}_{\text{Metric}} \leftarrow s\mathbf{D}_{\text{pred}} + b$
6:     **Mask Update:**
7:     $\mathcal{E} \leftarrow |\mathbf{D}_{\text{pred}} - \mathbf{D}_{\text{sensor}}| \odot \mathcal{M}$
8:     $\tau \leftarrow \mathcal{P}_{80}(\mathcal{E}[\mathcal{M} > 0])$
9:     $\mathcal{M} \leftarrow (\mathcal{E} < \tau) \odot \mathcal{M}$
10: **end for**
11: **return** $\mathbf{D}_{\text{Metric}}$

**Algorithm 2** Scale Fitting

**Require:** $\mathbf{D}_{\text{pred}}, \mathbf{D}_{\text{sensor}}, \mathcal{M}$
**Ensure:** $s, b \in \mathbb{R}$
1: **Extract valid pixels:**
$$\mathbf{p} = \mathbf{D}_{\text{pred}}[\mathcal{M}], \mathbf{s} = \mathbf{D}_{\text{sensor}}[\mathcal{M}]$$
2: **Solve:**
$$\min_{s,b} \|s\mathbf{p} + b\mathbf{1} - \mathbf{s}\|^2$$
3: **Solution:**
$$\begin{bmatrix} s \\ b \end{bmatrix} = \frac{1}{\Delta} \begin{bmatrix} N(\mathbf{p}^\top \mathbf{s}) - (\mathbf{1}^\top \mathbf{p})(\mathbf{1}^\top \mathbf{s}) \\ (\mathbf{p}^\top \mathbf{p})(\mathbf{1}^\top \mathbf{s}) - (\mathbf{p}^\top \mathbf{s})(\mathbf{1}^\top \mathbf{p}) \end{bmatrix}$$
$$\triangleright \text{ where } \Delta = N(\mathbf{p}^\top \mathbf{p}) - (\mathbf{1}^\top \mathbf{p})^2$$
4: **return** $s, b$

Figure 11: Depth Scale Alignment

**Normal Conditions** Surface normals capture fine geometric details and are scale-invariant. We compute per-frame normals using LOTUS (He et al., 2024). For datasets lacking multi-view RGB-D sensors, MoGe (Wang et al., 2025b) simultaneously estimates depth and normals, streamlining prelabeling.

### B.3 APPEARANCE CONDITIONS CONSTRUCTION

**Keyframe Selection.** Keyframes capture object and background appearance conditions. For simple tabletop tasks, the initial frame (all objects visible) serves as the object condition, and the final frame (objects removed) serves as the background condition. For complex tasks, an automated pipeline using VLM descriptors and Grounding DINO detects target objects, selecting the frame with the largest object pixel area for object conditions and the frame with the fewest for background conditions.

**Object Mask Generation.** Object descriptions, generated via structured prompts (Figure 12) specifying color, material, shape, and spatial position, are fed into Grounding DINO (Liu et al., 2024a) to generate bounding boxes, which SAM2 (Ravi et al., 2024) converts into object masks.

**Objects and Background Conditions.** Individual object patches are resized to $224 \times 224$ and passed through CLIP (Radford et al., 2021) to obtain embeddings. Background conditions are generated by masking out all detected objects and applying inpainting to reconstruct the object-free scene.

> **System Prompt for Object Descriptor:**
>
> You are an industrial robotic vision system with 100% detection guarantee. Perform comprehensive scene analysis to identify ALL movable objects, including occluded items. Strictly enforce size constraints and occlusion handling.
>
> Your goal is to complete a list of all visible objects on the table and process the description.
>
> **The output Template Format**:
> A [color] [object], [shape], located [region](x,y)
>
> **Examples:**
> - A red ball, spherical, at the center (250,300);
> - A brown chair, angular, in the top-left (100,50);
> - A silver-blue arm, mechanical, on the right (600,200);

Figure 12: Visual Description Prompt Template Architecture.

## C MORE QUALITATIVE RESULTS

### C.1 REAL-TO-REAL TRANSFER

*RoboTransfer* enables controllable generation of both foreground and background elements. Given the same structured input, the model allows flexible editing of background attributes such as texture and color (Figure 13) and foreground object appearance, including color (Figure 14). This real-to-real framework enriches training data diversity, improving policy generalization for downstream tasks.

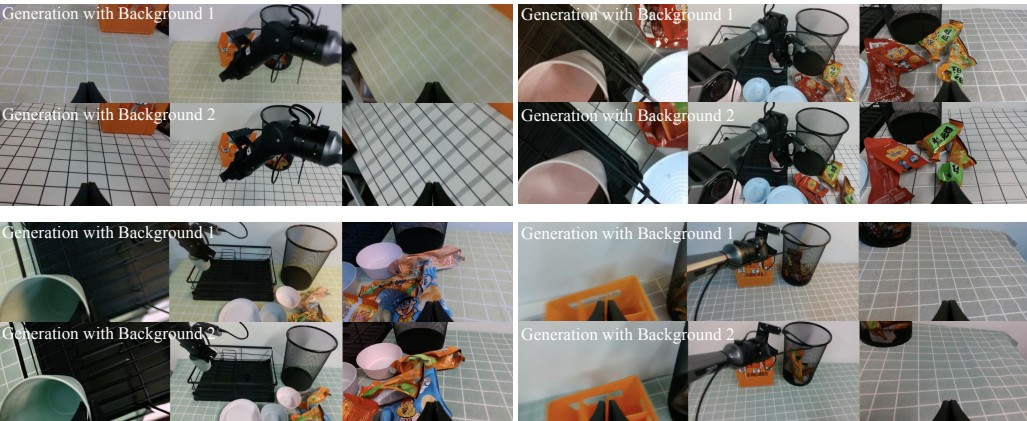

Figure 13: Visualizations of *RoboTransfer* with different background reference images.

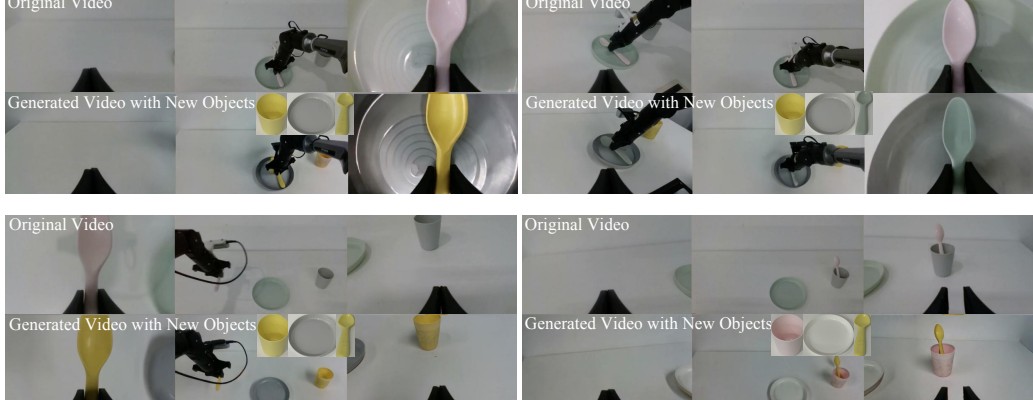

Figure 14: Visualizations of *RoboTransfer* with different object reference images.

## C.2 QUALITATIVE COMPARISON

Figure 15 compares robot data generation methods. Cosmos Transfer (Alhaija et al., 2025) performs well under fixed camera views but degrades notably for dynamic viewpoints. In contrast, *RoboTransfer* maintains strong multi-view consistency, producing realistic and coherent novel-view synthesis. RoboEngine (Yuan et al., 2025), based on image inpainting, suffers from noise and jitter, lacks temporal consistency, and cannot precisely control scene backgrounds or objects.

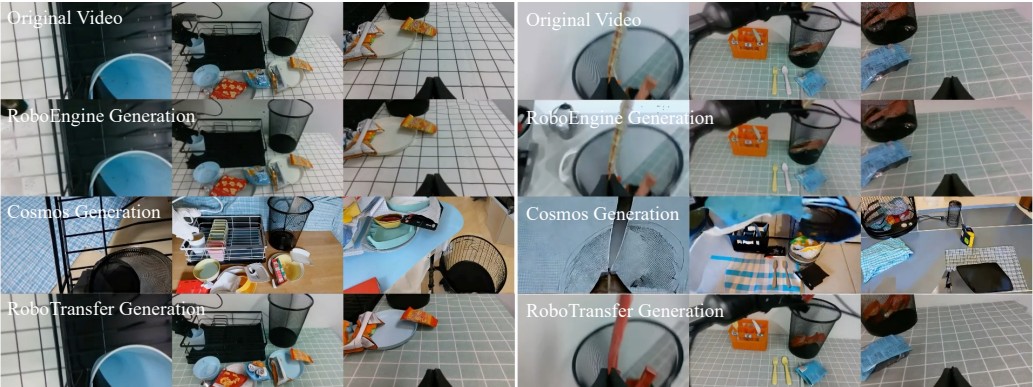

Figure 15: Comparison with Cosmos and RoboEngine generation results.

## C.3 DIVERSE SCENE SYNTHESIS

For fixed-arm tabletop tasks, backgrounds are simple, while mobile manipulation involves complex geometries. Figure 16 shows that *RoboTransfer* can generate richer, more diverse scenes in complex settings, enhancing the variety and realism of synthetic data for robotic learning.

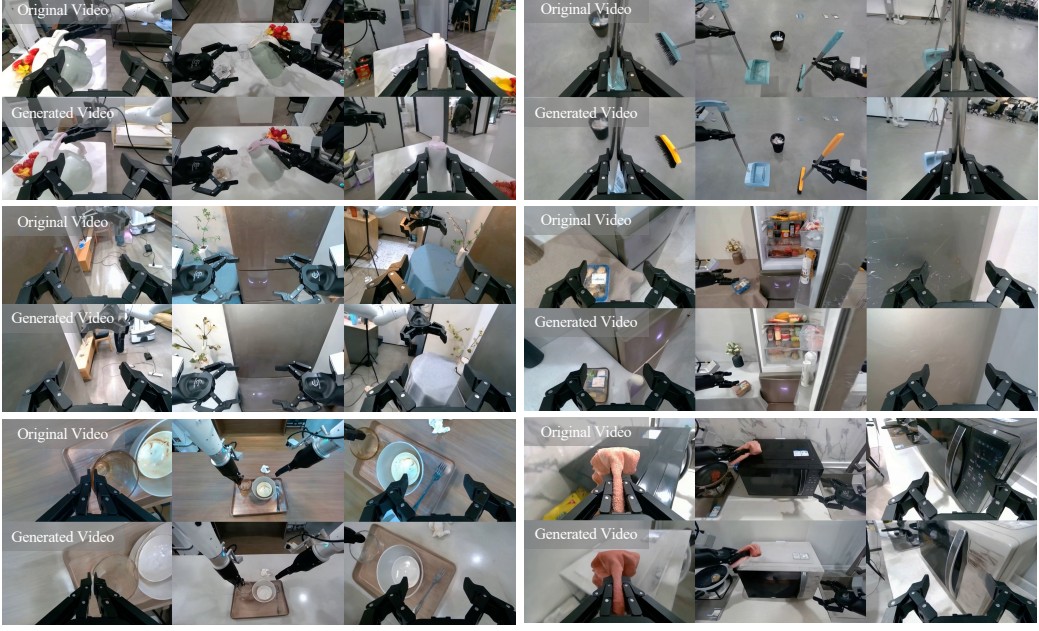

Figure 16: Visualizations of *RoboTransfer* for diversity scene generation results.

## D ROBOT POLICY MODEL IMPLEMENTATION DETAILS

To validate the effectiveness of the data generated by *RoboTransfer*, we train a visual policy using the procedures detailed below for both training and deployment.

### D.1 DATA COLLECTION AND PREPROCESSING

To ensure a fair evaluation, we excluded the *RoboTransfer* training dataset (Sec. B.1) from the real-robot experiments. Data preparation was conducted as follows:

**Real Expert Data** We collected 100 expert demonstration sets per manipulation task using the ALOHA teleoperation system. Observations included RGB images at 1280×720 resolution, downscaled to 640×360 for training efficiency, captured at 30Hz, and sampled at 10Hz. Auxiliary robot state information, including joint positions and end-effector poses, was recorded at 200Hz and downsampled to 50Hz for policy training.

**Synthetic Data Generation for Policy Fine-tuning.** To improve generalization, we generated synthetic videos based on the real demonstrations, introducing variations in foreground objects and background scenes. The pretrained diffusion model conditioned each synthetic video on: (1) per-frame 3D geometry inputs (depth and normal maps) from real demonstrations, and (2) a reference image containing novel objects and backgrounds from a held-out set independent of the policy training data.

### D.2 TRAINING PIPELINE AND CORE PARAMETERS

Our training pipeline uses the ACT (Action Chunking Transformer, Zhao et al. (2023)) architecture, processing visual input from three cameras (two wrist-mounted, one overhead). At each timestep, the model receives one RGB frame per view.

**Training Objective and Two-Stage Training Strategy.** The policy predicts the next 100 robot states (2s horizon at 50Hz). We adopt a pretrain-then-finetune strategy: 1) **Pretraining on Real Data:** The model was first pretrained for 100k steps using the collected real expert demonstration data. During this phase, the batch size was set to 512, and the learning rate was $1 \times 10^{-4}$. 2) **Finetuning with Synthetic Data:** After pretraining, the synthetic data was introduced to fine-tune the model for an additional 50k steps. The learning rate for this phase was reduced to $1 \times 10^{-5}$.

All training was performed on a cluster equipped with 8 NVIDIA H20 GPUs. The pretraining phase took approximately 24 hours, and the fine-tuning phase required about 12 hours.

### D.3 REAL-ROBOT DEPLOYMENT AND EVALUATION

**Deployment Platform.** Policies were evaluated on the Agilex Cobot Magic platform (Figure 9), the same system used for data collection to ensure consistency between training and evaluation environments.

**Inference Procedure** During deployment, the policy operates synchronously: the robot executes the full action sequence (100 actions) generated from the previous inference step before capturing new observations. At each decision point, the model receives one RGB frame per camera view along with the current robot state and outputs the next 100 actions. Inference latency is 10ms, and actions are executed at 50Hz, matching the training robot state frequency.

### D.4 COMPARISONS WITH EXTERNAL METHODS

To our knowledge, *RoboTransfer* is the first method for multi-view consistent data synthesis in robotic manipulation. Prior approaches exhibit clear multi-view inconsistencies and limited background control, as illustrated in C.2. While domain randomization handles object color changes, it cannot model localized or structurally meaningful variations. In contrast, *RoboTransfer* leverages a multi-view diffusion model to generate geometry-consistent, high-fidelity variations in both object appearance and background, supporting robust policy learning.

Table 5: Performance comparison across different test conditions and models. Success Rate (%) and Stage Score (0-4) are reported for each configuration.

| Model | Diff-Obj | | Diff-All | |
|---|---|---|---|---|
| | Success | Score | Success | Score |
| Baseline (Real only) | 33.3 | 2.7 | 13.3 | 1.6 |
| + Domain Random Aug | 44.4 | 2.89 | 11.1 | 1.58 |
| + Obj Aug | 44.4 | 3.0 | 22.2 | 2.0 |
| + Obj+Bg Aug | – | – | **46.7** | **3.0** |

## E  QUESTION AND ANSWER

### E.1  HOW TO DECOUPLE GEOMETRY AND APPEARANCE CONDITION?

In our framework, reference images serve primarily as appearance conditions, capturing visual properties like texture and color, rather than imposing strict geometric constraints. Global 3D geometry conditions ensure multi-view and temporal consistency across frames.

**1. Geometry conditions.** Multi-view depth and normal maps encode precise 3D structure and scene layout, even under dynamic wrist-mounted camera motion, defining where and how content should be rendered.

**2. Appearance transfer.** The reference image provides the target visual style (e.g., textures or background appearance) that the diffusion model transfers onto the global geometry. The reference does not need to be aligned with each frame; the model synthesizes a consistent appearance across views and time by using geometry as an anchor.

Qualitative results show visually coherent, realistic videos in wrist-view settings. Real-to-Real experiments confirm effective appearance transfer to dynamic robotic scenes, while Sim-to-Real experiments demonstrate that geometry from simulation can be faithfully combined with real-world appearance references. By decoupling geometry and appearance, our method improves robustness and generalization, requiring only diverse reference images rather than per-scene 3D scans or carefully aligned inputs, making it scalable for real-world deployment.

### E.2  THE COST OF SCALING MULTI-VIEW 3D DATA

Globally consistent 3D conditions are essential for wrist-view camera motion. While acquiring dense, high-quality 3D data can be resource-intensive, our framework leverages automated, cost-efficient pipelines:

**1. Real-robot data**: RGB-D cameras capture metric depth alongside RGB images. For both RGB-D and RGB-only images, depth and normal maps are robustly generated using automated pipelines (Figure 3). The code will be released to ensure reproducibility.

**2. Simulation data**: Per-view 3D geometry conditions (depth and normals) are trivially available at no extra cost, making simulation a scalable source of 3D-aware inputs.

By combining real and simulated pipelines, our framework reduces the effective cost of 3D data acquisition, enabling high diversity and fine-grained controllability in synthetic video generation for robotic policy learning.

### E.3  WHY NOT USE CLEAN BACKGROUND FRAMES AS REFERENCES?

RoboTransfer is designed for scalable, fully automated training. Capturing a clean background for every demonstration would be prohibitively expensive. Instead, background imagery is extracted from existing videos, preserving fidelity without extra data collection. In inference, a clean background frame can be captured if desired, keeping the process simple and efficient. This design balances automation, scalability, and flexibility across training and deployment.

## F    LLM Usage Statement

We used a large language model (LLM) to assist with the writing of this paper. Specifically, the LLM was utilized as a tool for grammar and spelling correction, as well as for refining sentence structure and improving overall readability.

## G    Ethics Statement

This work focuses on advancing data generation methods for robotic manipulation. The primary intended use is to improve sample efficiency and generalization in imitation learning. While synthetic data generation can reduce the need for large-scale real-world data collection, we acknowledge potential misuse of generative models, such as creating misleading or fabricated robotic demonstrations. To mitigate risks, we release our models and datasets solely for research purposes, under a permissive but non-commercial license. We further emphasize that this work does not involve human subjects, personal data, or sensitive content.

## H    Reproducibility Statement

We take reproducibility seriously and provide comprehensive details in the main text and supplementary material. Specifically:

The data processing pipeline, including depth/normal generation and scale alignment, is fully described (see Sec. B.1).

Training details, including architecture, hyperparameters, dataset splits, and compute requirements, are explicitly provided (see Sec. A.1).

We will release the full codebase, pretrained models, and processed datasets upon acceptance, enabling direct reproduction of our experiments.

Additional qualitative results, ablations, and instructions for dataset construction are included in the supplementary materials.

