# OpenReview forum: "RoboTransfer: Geometry-Consistent Video Diffusion for Robotic Visual Policy Transfer"
_ICLR.cc/2026/Conference — ICLR 2026 Conference Withdrawn Submission_

### Official Review · Reviewer_3QAL · 2025-10-28

**Soundness:** 2
**Presentation:** 3
**Contribution:** 2
**Rating:** 4
**Confidence:** 5

**Summary:**

The paper proposes RoboTransfer, a geometry-consistent video diffusion framework designed for robotic visual policy transfer. The method aims to expand real-world embodied datasets through controllable video generation, integrating multi-view information, depth maps, and surface normals to ensure geometric and temporal consistency. By decoupling geometry and appearance conditions (metric depth + normals vs. background + object references), the framework allows background or object-level visual editing while maintaining realistic robotic motion sequences. Experiments demonstrate improved geometric consistency and cross-view coherence, and show that mixing synthetic and real data improves downstream imitation-learning performance.

**Strengths:**

- The paper tackles an important bottleneck in robot imitation learning, namely the scarcity and limited diversity of real demonstration videos, by leveraging video diffusion for scalable data augmentation.

- The generation process introduces sufficient and disentangled control factors (geometry, background, and object-level appearance), resulting in multi-view and temporally consistent videos.

- Quantitative and qualitative results support that the generated videos are geometrically coherent and can improve real-world policy robustness under domain shifts.

**Weaknesses:**

- Lack of action-conditioned generation. Although the method enforces geometric and multi-view consistency via depth and normals, it does not explicitly condition the diffusion model on robotic action trajectories or policy states.
As a result, the generated videos are not guaranteed to be dynamically aligned with the original robot motions, potentially leading to mismatch between the visual sequence and the executed trajectory.

- The method is inherently constrained to cases where the object geometry and position remain approximately unchanged. Editing scenes with significant geometric variation (shape, pose, or spatial displacement) breaks the correspondence between the original action trajectory and the edited video. Consequently, the approach is mainly applicable to simple pick-and-place or rigid-body tasks, where conventional simulation-based augmentation already performs reasonably well and the sim-to-real gap is small. For complex interactions, e.g., deformable objects or articulated structures, RoboTransfer provides little advantage.

- The paper lacks discussion or quantitative comparison with recent action-conditioned video generation methods such as EVAC [1], which also employ multi-view geometry but directly condition on action sequences to ensure motion-video consistency.
Such comparisons would clarify the relative benefits of appearance-based vs. action-based conditioning.

[1]  EnerVerse-AC: Envisioning Embodied Environments with Action Condition

**Questions:**

- Could the authors integrate low-dimensional action embeddings or joint-state trajectories as additional conditions to improve motion fidelity?

- How sensitive is the generated video quality to errors in predicted depth or normal maps?

---

### Official Review · Reviewer_7D5G · 2025-10-31

**Soundness:** 3
**Presentation:** 2
**Contribution:** 2
**Rating:** 4
**Confidence:** 4

**Summary:**

The paper proposes a video data generation framework that takes a set of robotic manipulation data as input and generates more relevant but diversified data. The framework extracts the geometric and appearance conditions from the given videos and enforces them, as well as multi-view consistency, in the generated videos. Experiments show that robot policies trained with the generated data can achieve better performances compared to those trained with only the raw data.

**Strengths:**

- The paper addresses an important problem in data generation for robot learning, and it proposes a valid framework tackling that.
- Experiments show that the generated videos are not a naive duplication of the given data distribution, but do help improve the performance of the learned robot policy. The experiments also focus not only on video generation quality, but also show studies on its effects on robot learning, including real-robot experiments.

**Weaknesses:**

- A major concern of mine is that it seems there's no experimental comparisons with any baselines from other works, e.g., other video generation models, but just different ablated versions of the proposed model. I think the ablations are good justifications of the specific model components. But baseline comparisons can better justify the overall framework design from a higher level -- why this problem should be solved following this path, instead of some totally different paths? If such comparisons are hard to conduct, e.g., none of the existing works can tackle this specific problem without non-trivial adaptations, I think there should be a paragraph discussing and explaining this in Section 4.

- I feel the pipeline details, especially how the geometry and appearance conditions are extracted and incorporated, can be explained more clearly. For example, for multiple video keyframes and multiple backgrounds, the number of background and object inputs can both be >1, are their latent embeddings concatenated/aggregated? In addition to the text in Section 3.2, there could be more illustrations in Figure 2 and 3.

**Questions:**

- For the keyframe sampling (for extracting appearance conditions), how are they sampled? Is it a random sampling or is there a specific sampling strategy?

---

### Official Review · Reviewer_33qp · 2025-11-03

**Soundness:** 3
**Presentation:** 3
**Contribution:** 2
**Rating:** 4
**Confidence:** 4

**Summary:**

This paper proposes RoboTransfer, a diffusion-based video generative model for synthesizing robot data to boost imitation learning across diverse environments.
The key components of RoboTransfer are cross-view features interactions i.e., generates multiple views and leverages their cross-view interactions for improved consistency, and the use of 3D geometry (depth, normal maps)  when generating the robot data.
RoboTransfer qualitatively shows that it can generate videos with good geometric consistency and visual fidelity.
Experiments on real robot settings show that policies trained using the robot data generated from RoboTransfer generalize better to unseen scenarios.

**Strengths:**

- The proposed framework, RoboTransfer, is the first to propose a multi-view generation consistency in robotic data synthesis.

- The ability to offer fine-grained and disentangled control over the background attributes and the object attributes pose significant strengths in generating a diverse environment for the synthesized robot data.

- Thorough ablation studies demonstrate the efficacy of each design choice of RoboTransfer e.g., using predicted depth, using metric depth prediction together, using normal prediction together...

- Expeiments on real robot environments prove that the synthesized robot data from RobotTransfer significantly boosts the success rate on the Agilex Cobot Magic platform equipped with two PIPER robotic arms.

**Weaknesses:**

- Lack of algorithmic novelty. I believe that the main contribution of RoboTransfer lies in the consistent **multi-view** generation, which is facilitated by simply concatenating the multi-view frames along the width dimension. Conducting comparative experiments on this 'multi-view consistent modeling' scheme may have helped evidence the necessity and superiority of the given design, but the proposed modeling  scheme seems naive and straightforward.

- Lack of justification into why Metric D.P. and + N. leads to lower results compared to simple D.P. for head cameras unlike left/right cameras in Table 2.

- In the real-robot experiments, the synthetic data was generated based on real demonstrations with variations in foreground objects and background scecnes (L1038). As such, the provided RoboTransfer still requires expert demonstrations to acquire the true videos, and can only generate synthetic videos which adhere to the expert videos - this places a strong limit on the variety of videos RoboTransfer can generate.

**Questions:**

- Are there any hypothese into why the success rate / score declines temporarily at 80% ratio of synthetic data in Figure 8?

- Why does using Metric DP and N lead to lower results compared to simple DP in head cameras?

- What would the authors mention as the core technical contribution of their work?

- Am I correct in understanding that RoboTransfer can only generate synthetic robot data limited to the actions which exist in the expert demos?

---

### Official Review · Reviewer_G2VP · 2025-11-03

**Soundness:** 2
**Presentation:** 3
**Contribution:** 2
**Rating:** 2
**Confidence:** 4

**Summary:**

This paper presents a diffusion-based video generation framework designed for generating synthetic video data for training robot policies. The idea is to use reference images of backgrounds and objects and generate synthetic versions of real demonstration datasets for better generalizability of the manipulation policies. While the idea isn't new, this paper focusses on multi-view video generation (rather than just one view) with cross-view geometric consistency while providing fine-grained control over object and background appearance. The framework is evaluated in two types of experiments: video generation realism and its effect on the manipulation policy.

**Strengths:**

The paper aims to bridge the sim-to-real gap and overcome the limited-data challenge for real-world robot learning. The proposed geometry-aware diffusion pipeline is indeed a good contribution. Specifically, the explicit disentanglement of geometry and appearance for robotic video generation allows for finer control generation and with the multi-view generation can be particularly useful for robot learning. Often, robot learning policies use multi-view data (e.g., an environment camera and a wrist camera) but single view video generation may lead to un-aligned multi-view policies. This paper directly addresses that challenge.

**Weaknesses:**

While the overall idea is promising, the current version of the approach has several limitations.

1. The framework does not generate new motions or actions data. As far as I understand, it simply replaces the original videos with the generated ones while keeping the robot trajectory the same. Thus, the "synthetic" demonstrations still have actual controls, and just synthetic videos. Therefore, it is unclear to me how far from the real demos can it generalize. The changes in the background are such that they don't really affect the actual trajectory and the novel objects inserted have the same shape with just different colors. This to be severely limits the generalization.

2. On the empirical side, there were no comparisons with baselines in the main body of the paper. For video generation, the results only show ablations of the various losses constraint and on the policy learning side we see the same policy but with different levels of real/synthetic data. One would have expected to see comparisons with other baseline video generation and policy learning methods (including something like Pi0 or other VLAs that claim to be more inherently generalizable).

3. The experiments with multi-view generation are limited to fixed camera views and it is not clear how this would generalize to a new setup. For example, would we have to retrain the network for a new view?

**Questions:**

1. How different can the generated scenes be from the original demonstrations before the policy performance degrades? For instance, can RoboTransfer insert novel object geometries or rearranged spatial layouts, or is it restricted to appearance-level (texture/color) variations?

2. How does the model compare with recent video-diffusion or policy-learning baselines such as Cosmos-Transfer or VLA-based models (e.g., Pi0, OpenVLA)? Are there any quantitive performance metrics (besides just one qualitative example in Fig. 15)?

3. Since the method relies predicted metric depth and surface normals, how sensitive is performance to inaccuracies in these estimates?

---

### Note · Authors · 2025-11-12

I have read and agree with the venue's withdrawal policy on behalf of myself and my co-authors.